# Eicosanoids and Oxidative Stress in Diabetic Retinopathy

**DOI:** 10.3390/antiox9060520

**Published:** 2020-06-12

**Authors:** Mong-Heng Wang, George Hsiao, Mohamed Al-Shabrawey

**Affiliations:** 1Department of Physiology, Augusta University, Augusta, GA 30912, USA; 2Graduate Institute of Medical Sciences, College of Medicine, Taipei Medical University, Taipei 110, Taiwan; geohsiao@tmu.edu.tw; 3Department of Pharmacology, School of Medicine, College of Medicine, Taipei Medical University, Taipei 110, Taiwan; 4Department of Oral Biology and Diagnostic Sciences, Augusta University, Augusta, GA 30912, USA; 5Department of Cellular Biology and Anatomy, Augusta University, Augusta, GA 30912, USA; 6Culver Vision Discovery Institute and Ophthalmology, Augusta University, Augusta, GA 30912, USA

**Keywords:** eicosanoids, oxidative stress, diabetic retinopathy, cyclooxygenase, lipoxygenase, Cytochrome P450

## Abstract

Oxidative stress is an important factor to cause the pathogenesis of diabetic retinopathy (DR) because the retina has high vascularization and long-time light exposition. Cyclooxygenase (COX), lipoxygenase (LOX), and cytochrome P450 (CYP) enzymes can convert arachidonic acid (AA) into eicosanoids, which are important lipid mediators to regulate DR development. COX-derived metabolites appear to be significant factors causative to oxidative stress and retinal microvascular dysfunction. Several elegant studies have unraveled the importance of LOX-derived eicosanoids, including LTs and HETEs, to oxidative stress and retinal microvascular dysfunction. The role of CYP eicosanoids in DR is yet to be explored. There is clear evidence that CYP-derived epoxyeicosatrienoic acids (EETs) have detrimental effects on the retina. Our recent study showed that the renin-angiotensin system (RAS) activation augments retinal soluble epoxide hydrolase (sEH), a crucial enzyme degrading EETs. Our findings suggest that EETs blockade can enhance the ability of RAS blockade to prevent or mitigate microvascular damage in DR. This review will focus on the critical information related the function of these eicosanoids in the retina, the interaction between eicosanoids and reactive oxygen species (ROS), and the involvement of eicosanoids in DR. We also identify potential targets for the treatment of DR.

## 1. Introduction

Diabetes can be divided into type 1 (T1DM) and type 2 diabetes mellitus (T2DM). T1DM is mainly due to the autoimmune destruction of β cells [1,2]. Eventually, circulating insulin levels are negligible or completely absent in patients with T1DM [3]. T2DM is mainly associated with obesity, which affects one in three Americans [4,5]. Diabetic retinopathy (DR), a severe microvascular complication of T1DM and T2DM, is a disease that affects 7.7 million working-age adults in the U.S. DR-related blindness costs approximately $500 million annually in the U.S. [6] By 2050 more than a third of the U.S. population is expected to be diabetic; thus, the incidence of DR will increase dramatically [6]. The lack of productivity, high treatment costs, and diminished quality of life in patients with DR cause a pronounced socioeconomic burden.

In the retina, microvessels are vulnerable to oxidative stress because of chronic hyperglycemia, leading to increased reactive oxygen species (ROS) production. It has been observed that ROS imbalance is involved in DR [7]. The primary ROS species includes superoxide anion, hydroxyl anion, hydrogen peroxide (H_2_O_2_), and peroxynitrite [8]. Superoxide anion can be rapidly reduced to H_2_O_2_, which, due to its lipid-soluble properties, can modify cellular proteins, RNA, and DNA [9]. Hydroxide anion can oxidize DNA nucleotides and lipids. Superoxide anion may also react with NO to rapidly form peroxynitrite, which influences the properties of a variety of proteins, including inducible NO synthase (iNOS) and eNOS [10]. The endogenous antioxidant enzymes, including superoxide dismutase, glutathione peroxidase, and catalase, are involved in reducing these ROS. Of note, ROS play an essential role in the retinal pathological processes of DR, including inflammation and angiogenesis [11].

Since retinal microvascular dysfunction and damage are the key events in the onset and progression of DR, this review will focus on how arachidonic acid (AA; 20:4 n-6)-derived eicosanoids affect retinal microvascular function, and the role of these eicosanoids in DR. Our daily diets contain AA, which is the polyunsaturated fatty acid (PUFA). For example, AA is found in meat, including both red and white meat, organ meats, including kidney, liver, and brain, and eggs [12,13]. It is estimated that our mean daily AA intakes are about 100 to 350 mg in developed countries [12,13]. Additionally, humans can synthesize AA from linoleic acid (LA; 18:2 n-6), which is the principal PUFA in most western diets, including many nuts and seeds, vegetable oils, and products made from vegetable oils such as margarine [13]. AA is incorporated at the sn-2 position of the glycerol component of membrane phospholipids or other compound lipids. When the cell membrane is subjected to inflammatory stimuli, AA is released from the endogenous lipid pool by the action of phospholipase A2 (PLA2) (Figure 1). It is well-established that AA is converted by COX, LOX, and CYP pathway into eicosanoids (Figure 1) [14,15,16,17]. These lipid mediators can contribute considerably to oxidative stress, inflammation [15,18], and vascular function [19,20]. These three eicosanoid pathways are essential therapeutic targets for inflammatory and cardiovascular diseases because many receptors and metabolites of these three pathways are well defined. This review will provide valuable information related to the function of these lipid mediators in the retina and the involvement of these mediators in DR.

## 2. Functions of Eicosanoids in the Retina and Their Interaction with ROS

### 2.1. COX-Derived Eicosanoids

The major COX-derived eicosanoids include prostaglandins (PGs) and thromboxane (TX). COX enzymes catalyze the first two steps of the enzymatic reaction, including cyclooxygenase (dioxygenase) and peroxidase activity, to convert AA into PGH2 [21]. Of note, PGH2 is not stable, and it is the precursor for the production of PGs and TX, which is depending on the differential expression of isomerases and PG synthases in different tissues [22]. COX enzymes include two isozymes, COX-1 and COX-2. COX-1 is the constitutive isoform, which is responsible for the low PGs synthesis required for cell homeostasis, whereas COX-2 is inducible by many extracellular stimuli, including cytokines and growth factors, during chronic inflammation [17,23]. The major PGs are PGD2, PGE2, PGI2, and PGF2α, and the central TX is TXA2. The function of these PGs and TXA2 is mediated via the binding of DP, EPs (EP1 to EP4), IP, FP, and TP receptors.

It is well established that PGs are produced in retinal and choroidal blood vessels [24,25,26]. Several studies [27,28] have demonstrated that PGs play a vital role in the regulation of retinal blood flow (RBF) and choroidal blood flow (ChBF) [27,28]. Of note, RBF supplies the inner layers of the retina, whereas ChBF nourishes the outer layers (retinal pigment epithelium (RPE) and the photoreceptors) of the retina with nutrition and oxygen. During a rise in perfusion pressure, PGE2 and PGF2α are amply released in retinal and choroidal blood vessels, and these PGs cause vasoconstriction [29]. On the other hand, the release of PGI2 and PGD2 in retinal and choroidal blood vessels during perfusion causes vasorelaxation [29]. Notably, several studies [27,30] have suggested that PGs are involved in the inability of autoregulation of RBF and ChBF in newborn animals, which results in excess delivery of oxygen to the retina, and causes retinal microvascular damage in retinopathy of prematurity (ROP). Moreover, PGE2 has detrimental effects on the blood vessels, including increased oxidative stress, increased vasodilation, increased vascular permeability, and increased production of proinflammatory cytokines [16].

As mentioned above, PGs are produced from AA released from the phospholipids of the cell membrane by COXs by generating PGG2 and PGH2. The reaction for the generation of PGG2 by COXs needs peroxides as critical components in COX activation. Thus, several studies [31,32,33] have reported that in the blood vessels, ROS can affect both the activity and expression of COXs. For example, H_2_O_2_ increases COX-2 expression in endothelial or vascular smooth muscle cells (VSMCs) [31,32,33]. Moreover, there is some evidence that PGs can directly regulate the production of ROS in the blood vessels. For example, TXA2 up-regulates the expression and activity of NADPH oxidase, a crucial enzyme to produce ROS in all blood vessel wall cells [34]. PGE2 promotes ROS formation via the EP1 receptor, which is associated with hypertension and endothelial dysfunction [35,36].

### 2.2. LOX-Derived Eicosanoids

The major components in LOX-derived eicosanoids are hydroxyeicosatetraenoic acids (HETEs), including 5-HETE, 8-HETE, 12-HETE, and 15-HETE, as well as leukotrienes (LTs), containing LTA4, LTB4, LTC4, LTD4, LTE4, and LTF4 (Figure 1). Under the catalysis of lipoxygenases (5-LOX, 8-LOX, 12-LOX, and 15-LOX), which are non-heme iron-containing enzymes, AA is metabolized into hydroperoxyeicosatetraenoic acid (HpETE). 12-HpETE and 15-HpETE are reduced into 12-HETE and 15-HETE. 5-HETE is generated from 5-LOX, and 8-HETE is generated from 8-LOX. 5-HETE induces the degranulation of neutrophils, and 8-HETE is involved in skin inflammation. 5-LOX metabolizes AA into 5-HpETE, which is the precursor for the synthesis of proinflammatory LTA4. LTA4 is metabolized to LTB4 by leukotriene A4 hydrolase, and LTB4 binds to its receptors (BLT1 or BLT2) for its action [37]. LTA4 is unstable, and it can be merged with glutathione to form cysteinyl-LTs (cysLTs), which include LTC4, LTD4, LTE4, and LTF4 [38,39]. The function of LTs is mediated through the binding of these LTs to their receptors, including BLT, cysLT, and LTE4 receptors.

LOX-derived eicosanoids are implicated in several critical inflammatory conditions, and LTs (LTC4, LTD4, and LTE4) are mostly synthesized by neutrophils, macrophages, and mast cells [38]. As LTs have a relatively short half-life, these lipid mediators act as autacoids near their synthesizing sites [40]. LTC4, LTD4, and LTE4 were named slow-reacting substances of anaphylaxis because these products cause contractions in the smooth muscles of guinea pig ileum [41]. A substantial body of evidence indicates that LTB4 promotes leukocyte chemotaxis, adhesion and degranulation, and enhances oxidative stress, vascular permeability, and the production of proinflammatory cytokines [40,42]. 12-HETE and 15-HETE are generated in the microvessels. Several studies [43,44] indicate that 12-HETE is a potent vasodilator, whereas 15-HETE causes vasoconstriction. Notably, 5-LOX-derived products act as potent chemotactic agents for the recruitment of several proinflammatory cells, namely neutrophils, eosinophils, and monocytes in the blood vessels [40,45]. Accordingly, most investigators consider 5-LOX products to be detrimental factors in pathological conditions, including asthma [41], rheumatoid arthritis [41], DR [46], and cardiovascular diseases [40,45]. Leukotriene antagonists are clinically beneficial to treat asthma because leukotriene induces bronchoconstriction [41].

Although the mechanisms that LOX-derived metabolites promote ROS have not been well established, these metabolites are reported to act as the upstream of NADPH oxidase (NOX) pathways in cancer research [47]. For example, 12-HETE stimulates NOX1-mediated ROS production and migration in colon adenocarcinoma cells [48]. Besides 12-HETE, some evidence suggests that 15-HETE induces apoptosis in K-562 cells (myeloid leukemia) through NOX-mediated ROS production [49]. Of note, several studies have reported that BLT2, the LTB4 receptor, is involved in cancer cell growth and proliferation via NOX-mediated ROS production [50,51].

### 2.3. CYP-Derived Eicosanoids

The CYP-derived eicosanoids contain epoxyeicosatrienoic acids (EETs) (5,6-EET, 8,9-EET, 11,12-EET, and 14,15-EET) and 20-HETE (Figure 1). The CYP enzymes, about 45–55 kDa, are heme-containing proteins [52]. In the presence of NADPH and oxygen, AA is oxidized by the CYP enzyme system into EETs and HETEs. Among HETEs, 20-HETE, the ω-hydroxylation product of AA, is the major lipid metabolite in blood vessels and the kidneys [53]. CYP4A and CYP4F isoforms are the major enzymes for 20-HETE synthesis [16]. Production of EETs is less specific, and several CYP isoforms are involved, including CYP1A, 2B, 2C, 2D, 2E, and 2J [16]. It is well established that the major CYP epoxygenases for EETs synthesis in the kidneys and the microvasculature are CYP2C and CYP2J [15]. For example, CYP2C11, a rat isoform, has the highest epoxygenase activity, whereas CYP2C24 has the lowest activity [15]. Similarly, CYP2J isoforms are involved in EETs synthesis [54,55]. These CYP-derived eicosanoids are essential to regulate cardiovascular and renal function [15,16]. 20-HETE and EETs have been connected to regulate vascular function [15]. 20-HETE causes vasoconstriction in the microvasculature, and it is a vital regulator of the myogenic tone [17]. Notably, EETs elicit relaxation in the microvasculature, and they are the endothelium-derived hyperpolarizing factors [56,57]. Importantly, EETs can act as angiogenic factors [15,58]. EETs are readily hydrolyzed by soluble epoxide hydrolase (sEH) to form 5,6-dihydroxyepoxyeicosatrienoic acids (DHETs), 8,9-DHET, 11,12-DHET, and 14,15-DHET, which possess less biologically active than are EETs [15,58]. It is well established that sEH is the crucial enzyme in the metabolism of EETs, and blockades and deletion of sEH can change the level of EETs in vivo [15,58].

It has been demonstrated that EETs are generated in the retina [59], retinal endothelial cells [60], and significant EETs levels are evident in vitreous samples from diabetic patients [61]. Although it has been demonstrated that CYP2C and CYP2J isoforms are responsible for EET production in the kidneys [62,63], the major CYP enzymes for retinal EETs production are still unclear. Our recent study [64] has determined the expression of retinal CYP2C and CYP2J in mice compared with renal tissues as a positive control. We showed that CYP2J expression is absent, and CYP2C isoforms are the major epoxygenases in mouse retina [64]. Interestingly, these findings suggest that the blockade of CYP2C could be a target to reduce ROS production in the diabetic retina because CYP2C generates detrimental ROS in the heart [65]. Notably, a recent study by Park et al. has demonstrated that GPR40 is an EET receptor in vascular cells, and GPR40 plays a role in endothelial proliferation and tube formation that can contribute to angiogenesis [66]. To address whether EET receptor (GPR40) is expressed in the retina, we determined the expression of GPR40 in retinal samples isolated from mice and in human retinal endothelial cells (HRECs). The animal protocol was approved by the Institutional Animal Care and Use Committee. It was in accord with the requirements of the National Research Council Guide for the Care and Use of Laboratory Animals. Using the pancreas as a positive control [66], the expression of GPR40 is shown in the mouse retina and HRECs (Figure 2A). Notably, GPR40 is also highly expressed in the retinal blood vessels (Figure 2B). These results suggest that GPR40 could be a target to modulate the action of EETs in the retina. To determine the physiological function of 20-HETE and EETs in the retina, Metea et al. [67] investigated the role of HET0016 (a 20-HETE-selective blocker [53]) and MS-PPOH (an EETs-selective blocker [53]) in the function of retinal arterioles. This study [67] showed that glial-evoked vasodilation was blocked by MS-PPOH, whereas HET0016 blocked vasoconstriction. These results support the hypothesis that glial stimulation elicits vasodilation via EETs, whereas glial stimulation results in vasoconstriction mediated by 20-HETE in the retina.

As mentioned above, CYP enzymes are involved in the synthesis of EETs and 20-HETE. Several studies [68,69] suggested that CYP catalytic cycle’s poor coupling results in the continuous production of ROS, which affects different signaling pathways and other cellular functions. Edin et al. [69] determined the effects of increased endothelial expression of CYP2C8 (Tie2-CYP2C8) and CYP2J2 (Tie2-CYP2J2) transgenic mice to ischemia/reperfusion (I/R) injury in the isolated heart. They showed that infarct size was unchanged in Tie2-CYP2J2 mice after I/R, whereas Tie2-CYP2C8 mouse hearts had significantly increased infarct size after I/R. The reason for increasing the infarct size of Tie2-CYP2C8 hearts is because of increased ROS production. These results support the notion that CYP2J2 generates cardioprotective EETs, whereas another isozyme in the heart, CYP2C, generates EETs as well as detrimental ROS [65]. Some evidence suggests that hepatic fibrosis may be mediated in liver disease is through CYP2E1-dependent release of ROS from hepatocytes, which then may stimulate collagen production in stellate cells [70].

## 3. Diabetic Retinopathy

The retina is an extension of the central nervous system and highly metabolic active organ. It is made of several layers, and the retinal layers from inside to outside are nerve fiber (NF), ganglion cell (GC), inner plexiform (IPL), inner nuclear (INL), outer plexiform (OPL), outer nuclear (ONL), and RPE (Figure 3A) [11]. The retina is related anteriorly to the vitreous, lens, and cornea, which are avascular transparent media. The development of abnormal blood vessels in the vitreous such as in DR causes interruption of light and vision deterioration. Retinal vessels are localized in the inner neural retina, where they are distributed in the nerve fiber, inner and outer plexiform layers (Figure 3B) [11]. However, the photoreceptor layer lacks retinal vessels and receives its nutrition by diffusion from choroidal blood vessels. The flow of nutrients materials, metabolites, ions, proteins, and water flux to and from the retina is regulated by two blood-retinal barriers (BRB), inner and outer. The inner barrier is made of endothelial cells, pericytes, and glial cells; however, the outer barrier is made of RPE and choroidal endothelial cells. Disruption of inner and outer barriers is a characteristic feature of retinal diseases such as DR and age-related macular degeneration.

DR, a neurovascular complication, remains one of the most common causes of blindness worldwide. World Health Organization (WHO) has placed DR on the top list of eye conditions that should be treated [11,71]. The microvascular dysfunction in DR is characterized by apoptosis of retinal pericytes and endothelial cells, leading to BRB breakdown, capillary degeneration, and development of retinal ischemia. BRB breakdown causes retinal hyperpermeability and development of diabetic macular edema (DME), which is a leading cause of vision loss in DR. Capillary degeneration leads to the development of relative retinal ischemia and subsequently VEGF-dependent retinal neovascularization, a cardinal sign of proliferative DR (PDR). Current methods for treating diabetic retinopathy (DR), including laser photocoagulation and anti-vascular endothelial growth factor (VEGF), are limited by significant side effects and do not eliminate the risk of blindness. Thus, there is a critical need to identify new therapeutic targets for the treatment of DR [72,73].

The retina comprises a high content of PUFA, and it also has high oxygen and glucose uptake as compared with other tissues. Thus, the retina is more prone to oxidative stress [11]. Chronic hyperglycemia is related to the development of DR, although the underlying mechanisms of this association are still not clear. Several biochemical pathways and molecular mechanisms have been implicated in the possible links, including activation of the renin-angiotensin system (RAS), increased advanced glycation end products, dysregulation of the polyol pathway, activation of PKC, and chronic inflammation [11]. Of note, many of these pathways are leading to ROS production and the burden of oxidative stress in retinal tissues [11]. DR is mostly a disease of the retinal microvasculature, although damage to neurons and glia also occurs [11]. This review will focus on the role of eicosanoids in oxidative stress, retinal microvascular dysfunction, and neovascularization to DR. The following section will provide the reader with a clear summary of the role of eicosanoids and omega 3 (ω-3)-derived metabolites in DR.

## 4. Eicosanoids and Diabetic Retinopathy

### 4.1. Role of COX-Eicosanoids in Diabetic Retinopathy

In DR, vascular leakiness and proliferation are two important factors to cause vision impairment [74,75]. Patients with diabetes are usually vulnerable to DR, and retinal neovascularization (NV) in the late stage of DR could lead to blindness. Of note, in DR, ischemia is the common precursor to NV, and it is well established that early proinflammatory genes are generally expressed in ischemic retina [76]. One of these genes expressed at high levels in the early stages of DR is COX-2, which is induced by inflammatory cytokines [16,17]. Moreover, increased PGs in DR have been found in the vitreous cavity in both animal and clinical studies [77,78]. Thus, a lot of research effort has focused on the role of COX-2 and PGE2 in the pathogenesis of DR. VEGF is a proinflammatory molecule that plays an essential role in the development of vascular leakage and retinal NV in DR [79]. A previous study [80] has shown that PGE2 increases VEGF expression in cultured Müller cells. Interestingly, Yanni et al. [81] have reported that in Müller cells, activation of the EP4 receptor, the receptor of PGE2, increases VEGF production. In contrast, a blockade of EP4 receptors decreases VEGF production in a concentration-dependent manner. An EP4 blockade by L-161982 significantly reduced pathologic NV in oxygen-induced retinopathy (OIR). Noteworthy, earlier work by Ayalasomayajula et al. [82] has demonstrated that celecoxib (a selective COX-2 inhibitor) inhibited VEGF expression without any significant effect in COX-2 expression. Moreover, the COX-2 blockade significantly decreased vitreous to plasma protein ratio, which is an index of the retinal vascular leakage in diabetic rats. These results support the hypothesis that COX-2 and EP4 could be valuable therapeutic targets for the early stages of DR and proliferative DR.

NF-κB is a family of highly conserved transcription factors that regulate many genes involved in the inflammatory response [79]. Thus, NF-κB is a proinflammatory transcription factor. NF-κB is composed of homodimers and heterodimers, and the most abundant forms are the p65 and p50 subunits [79]. The NF-κB proteins are typically sequestered in the cytoplasm by IκB. The primary mechanism for NF-κB activation is the inducible degradation of IκB triggered through its site-specific phosphorylation by the IκB kinase (IKK) complex, resulting in IκB degradation [79]. The degradation of IκB releases the NF-κB heterodimers to translocate to the nucleus where they bind to nuclear DNA, leading to activation of inflammatory mediators, including tumor necrosis factor α (TNFα), ICAM-1, and interleukin-1β. Of note, NF-κB activation induces inflammatory mediators and increases oxidative stress, which is involved in the pathogenesis of DR [83,84,85,86]. Notably, Zheng et al. [87] have reported that treatment with aspirin (a COX inhibitor) not only inhibited NF-κB activation, but also inhibited the expression of iNOS, ICAM-1, VCAM, and capillary degeneration and capillary cell death in the diabetic retina. Moreover, NF-κB activation contributes to ROS generation [11]. These results provide substantial evidence that aspirin-mediated inhibition of capillary degeneration in the early stage of DR is mediated via inhibition of NF-κB and the subsequent oxidative stress and inflammatory response.

Several studies have determined the effects of COX blockade in clinical studies [88,89,90,91]. In the Early Treatment Diabetic Retinopathy Study (ETDRS), researchers examined the effects of aspirin (650 mg per day) or placebo in 3711 patients with mild-to-severe nonproliferative DR (NPDR) or early proliferative DR (PDR) [88]. They found that aspirin did not prevent high-risk PDR development and did not reduce the risk of visual loss in these DR patients. This study suggests that the COX blockade does not have beneficial effects in advanced DR patients. The reason that aspirin did not provide protective effects in advanced DR patients is still not clear. It could be due to the dose of aspirin because, in a previous study, Zheng et al. [87] used the dose of 26 mg/kg/day, which is about 1820 mg per day based on the bodyweight of 70 kg of healthy persons, to inhibit the early lesion of DR in diabetic rats. Thus, a higher dose of aspirin is needed to provide beneficial effects in DR patients.

Interestingly, in the Dipyridamole Aspirin Microangiopathy Diabetes Study (DAMAD) trial, a higher dose of aspirin (990 mg per day) has a significant protective effect of slowing the development of retinal microaneurysms [89]. Moreover, a pilot study in Japan [90], researchers investigated the effects of sulindac (a non-specific COX-2 inhibitor; 200 mg/day, 100 mg twice a day; *n* = 16) on DR progression in patients with T2DM as compared to controls (24 patients) for three years. They found that patients in the sulindac group did not develop DR, nor was there the progression of pathology in those who began the study with mild NPDR. On the other hand, six patients progressed to mild NPDR in the control group. Subsequently, a prospective randomized study [91] showed that treatment with celecoxib caused the reduction of fluorescein leakage in patients with diabetic macular edema. These clinical studies [88,89,90,91] support the notion that COX blockade might have beneficial effects in the development of DR in patients with the early stages of DR.

### 4.2. Role of LOX-Eicosanoids in Diabetic Retinopathy

It is well established that inflammatory insults to the retina are essential factors in the development of the early stages of DR. 5-LOX-derived metabolites, including LTB4 and cysteinyl leukotrienes (LTC4, LTD4, and LTE4), play an important role in the inflammatory processes. To determine the role of 5-LOX in the pathogenesis of DR, a study by Gubitosi-Klug et al. [37] has evaluated the role of 5-LOX knockout (KO) and wild-type (WT) mice in the development of DR. They found that diabetic WT mice developed degeneration of retinal capillaries and pericyte at nine months post-streptozotocin (STZ) treatment and increases in both leukostasis and superoxide production at three months post-STZ treatment. Diabetic 5-LOX KO mice developed less capillary degeneration and loss of pericytes and less leukostasis, less superoxide production, and less activation of NF-κB, which contributes to ROS generation [11]. These results provide substantial evidence that 5-LOX-derived metabolites promote proinflammatory mediators and oxidative stress, and play a role in DR’s pathogenesis. Moreover, using STZ-induced diabetic mouse model and cultured retinal cells, Talahalli et al. [46] showed that bone marrow-derived cells from diabetic mice synthesize more LTB4 than do those from WT mice; the mouse retina, retinal glial cells, and retinal endothelial cells (mREC) need LTA4 for the synthesis of LTB4 by transcellular metabolism; and high-glucose conditions increase BLT1 receptor expression in retinal glial cells and mREC, which then cause retinal microvascular endothelial cell death. These results support the notion that transcellular delivery of LTA4 from bone marrow-derived cells to retinal cells results in the production of LTB4, which can contribute to chronic inflammation and the development of DR. Retinal angiogenesis, the formation of new blood vessels in the retinal vasculature, is one the most damaging pathological events occurring during advanced DR. Several studies [92,93,94] have demonstrated that 5-LOX-derived leukotrienes promote retinal NF-κB expression and its subsequent downstream target pathways, including ROS production, cytokine molecules, and adhesion molecules; these lipid mediators cause leukostasis and degeneration of retinal capillaries; they increase retinal microvascular permeability leading to retinal edema; they activate NADPH oxidase, thereby increasing oxidative stress; and they promote retinal endothelial cell proliferation and migration. All of these pathological actions can contribute to the retinal angiogenesis in advanced DR.

Extensive research activities have focused on the effect of 12/15-LOX products in the pathogenesis of DM. 12/15-LOX is found in the retinal cells, including endothelial cells and glial cells [95]. The primary 12/15-LOX-derived products from AA are 12-HETE and 15-HETE. 12-HETE has a role in various biological processes, including atherogenesis, cancer cell growth, and neuronal apoptosis. Moreover, 12-HETE has proinflammatory effects [96,97] and has been implicated in diabetic vascular complications [98]. To determine the role of 12/15-LOX in PDR, we showed that 12-HETE and 15-HETE production were significantly increased in oxygen-induced ischemic retinopathy (OIR), a model of PDR [95]. We then found that 12-HETE and 15-HEHE levels were elevated in the vitreous of diabetic patients with PDR. Interestingly, the blockade or deletion of 12/15-LOX attenuated retinal NV. Additionally, 12-HETE administration augmented VEGF expression in Müller cells and astrocytes. These results support the hypothesis that 12-HETE and 15-HETE production by 12/15-LOX are essential regulators of retinal NV through modulation of VEGF expression and could provide a new therapeutic target to prevent and treat ischemic retinopathy. Our research group then determined the effect of 12/15-LOX metabolites on endothelial cell barrier function in the presence or absence of NADPH oxidase, an important enzyme to produce ROS, inhibitors. Our previous study [99] showed that activation of 12/15-LOX is a contributing factor to the vascular hyperpermeability during DR and that NADPH oxidase plays a role in this process via activating VEGF receptor 2 (VEGF-R2)-signal pathway. Interestingly, our another study [100] showed that the products of the 12/15-LOX pathway were significantly up-regulated under hyperglycemic conditions with 15-HETE exhibiting the most significant increase; 15-HETE activates retinal endothelial cells through the NOX system leading to increases in leukocyte adhesion, hyperpermeability, and finally NV, the cardinal signs of DR. Based on these previous studies [95,99,100], we proposed a hypothesis that hyperglycemia activates PLA2 to release AA from the retinal cell membrane. AA is then converted to 12-HETE or 15-HETE that generates ROS through NOX, creating a status of oxidative stress. Oxidative stress leads to the activation of retinal endothelial cells through various inflammatory signaling pathways, leading to leukocyte adhesion, hyperpermeability, and ultimately NV, which is the pathogenesis of advanced DR (Figure 4).

To study the biological effects of 12/15-LOX in DR, our previous study [99] determined the effects of baicalein, 12/15-LOX inhibitor, in diabetic mice. Treatment of diabetic mice with baicalein significantly decreased retinal HETE, intercellular adhesion molecule 1 (ICAM-1), vascular cell adhesion molecule 1 (VCAM-1), interleukin 6 (IL-6), ROS generation, and NOX2 expression. Baicalein also reduced VEGF-R2 levels in the diabetic retina. Our findings suggest that 12/15-LOX contributes to vascular hyperpermeability during DR via the NADPH oxidase-dependent mechanism, which involves the suppression of protein tyrosine phosphatase and activation of VEGF-R2 signal pathway [99]. Besides baicalein, several pharmacological inhibitors, including nordihydroguaiaretic acid (NDGA), cinnamyl-3,4-dihydroxy-cyanocinnamate (CDC), have been developed. However, these 12/15-LOX inhibitors did not show a clear isoform specificity [101,102]. Moreover, these inhibitors display a species-specificity, and they have off-target effects. For example, these inhibitors have anti-oxidative properties, and they affect the cellular redox homeostasis. Thus, it is hard to determine which of the two functions, LOX inhibition or redox homeostasis, is the main reason for the detected biological result [103]. Consequently, results obtained from these inhibitors must be interpreted with caution. To address this issue, the inhibitor studies should always be confirmed by another approach, for example, 12/15-LOX KO [11].

### 4.3. Role of CYP-Eicosanoids in Diabetic Retinopathy

As compared to COX and LOX pathways, the role of the CYP pathway in retinopathy and DR is less documented. Although it has been reported that EETs are involved in neurovascular coupling [67], little is known about retinal angiogenesis. A previous publication by Michaelis et al. [60] is the first to determine the role of CYP2C-derived EETs in hypoxia-induced cell migration and angiogenesis. They showed that CYP2C isoforms are expressed, and EETs are generated in cultured retinal endothelial cells. Additionally, hypoxia-induced CYP2C protein expression and EET formation. Moreover, the CYP2C blockade attenuated the effects of EETs on endothelial cell migration and endothelial tube formation. These results support the notion that endothelial EETs are implicated in retinal angiogenesis, especially under hypoxia conditions. Future study is needed to investigate whether CYP2C-induced ROS production [65] contributes to retinal angiogenesis under hypoxia. Importantly, another previous study showed that 11,12-EET has a proangiogenic activity in the retina following hypoxia [104], which supports the notion that EETs blockade could be a therapeutic target for retinal NV. EETs have been shown to promote retinal NV in OIR [105], and the CYP2C blockade provides the protective effects on pathological retinal NV in OIR [106]. To determine lipidomic profiles of various PUFA, which including LA, AA, eicosapentaenoic acid (EPA, 20:5 n-3), and docosahexaenoic acid (DHA, 22:6 n-3), we used LC/MS/MS to measure the levels of 12/15-LOX-, COX-, and CYP-derived metabolites in diabetic (STZ model) and control mice. Among the 107 lipid metabolites screened, only a few lipids were significantly increased in diabetic mice. Notably, we found that 5,6-DHET, 11,12-DHET, and 14,15-DHET levels are significantly elevated in diabetic mice [107]. These results suggest that retinal sEH levels and activity are elevated in diabetes.

A robust body of literature has established the role of RAS in DR [108,109,110,111,112,113,114,115]. In RAS, prorenin is activated to form renin [108,109,110,111,112], which converts angiotensinogen to Angiotensin I (Ang I) [108,109,116]. Ang I is then hydrolyzed by angiotensin-converting enzyme (ACE) to produce Angiotensin II (Ang II) [108,109]. Ang II is the major bioactive product of RAS, and Ang II receptor type 1 (AT1 receptor) is the primary receptor to mediate the function of Ang II [117,118,119,120]. Several clinical studies are designed to determine the effects of RAS blockade in the development of DR because RAS activation is implicated in DR development in animal studies [108,109,110,111,112]. EUCLID trial demonstrates that ACE blockade by lisinopril attenuated the progress of DR to proliferative DR [121]. The DIRECT trial was divided into the DIRECT-Prevent group (*n* > 1400) and the DIRECT-Protect group (*n* > 1900) [108,122]. While AT1 blockade by candesartan decreased the incidence of DR, AT1 blockade did not attenuate the progression of established DR [122]. Thus, a lack of understanding of the molecular mechanism of retinal microvascular damage induced by RAS is a critical barrier to the use of RAS blockade to prevent or treat DR. To address the disappointing results of the DIRECT trial, our recent study [64] shows that Ang II increases retinal sEH expression, which is blunted by an AT1 blocker; 11,12-EET exacerbates Ang II-retinal vascular leakage; diabetes (STZ model) induces retinal angiotensinogen and AT1 expression (RAS activation), which is associated with increased retinal sEH expression; and sEH KO (increasing EETs) exacerbates diabetes-induced retinal vascular leakage. Based on these results, we propose a hypothesis that during diabetes, RAS activation augments retinal sEH, via AT1, which decreases EETs (pro-permeability and pro-angiogenesis factors) to counterbalance the effects of RAS on retinal microvascular damage (Figure 5).

### 4.4. Role of Omega 3-Derived Metabolites in Diabetic Retinopathy

Besides generating from omega 6 (ω-6) PUFAs, such as AA, eicosanoids can also be produced from ω-3 PUFA, such as EPA and DHA. Several studies have reported that COX- and LOX-derived products generated from ω-3 PUFAs inhibit inflammation and angiogenesis, and may possess protective effects in the development of retinopathy and DR [105,123]. To determine the role of ω-3 PUFA COX-derived PGs on angiogenesis, Szymczak et al. investigated the modulation of proangiogenic activation of human endothelial cells (ECs) by ω-3 PUFA [123]. They showed that ω-6 PUFAs stimulate, but ω-3 PUFAs inhibit major proangiogenic processes in human ECs, including the induction of angiopoietin-2 (Ang2), endothelial invasion, and tube formation, that are usually activated by the major ω-6 PUFA AA [123]. Importantly, they found that PGE3 (ω-3 PUFA-derived PG) suppressed the induction of Ang2, a vital factor in the angiogenic differentiation of ECs, by growth factors in human ECs, which is the opposite to the effects of PGE2. These results support the notion that ω-3 PUFA-derived PG protects against NV as compared with the detrimental effects of ω-6 PUFA-derived PG. Furthermore, resolvins, produced from ω-3 PUFAs via COX-2, have potent anti-inflammatory properties by blocking the production of pro-inflammatory factors [124]. To determine the role of ω-3 PUFA-derived metabolites in NV, Sapieha et al. [125] fed ω-3 PUFA diets into COX-1 KO, COX-2 KO, 5-LOX KO, 12/15-LOX KO, and WT mice, and then retinopathy was induced by oxygen exposure (OIR). They showed that only 5-LOX KO mice, but not COX-1 KO, COX-2 KO, or 12/15-LOX KO, abrogated the protection against OIR by dietary ω-3 PUFAs. This study identified 4-hydroxydocosahexaenoic acid (4-HDHA), a product of ω-3 PUFA via 5-LOX, which regulates NV in OIR modeling ROP and advanced DR [125].

To determine the role of ω-3 and ω-6 PUFA-derived metabolites via CYP in NV, Shao et al. [126] fed ω-3 or ω-6 PUFA diets into Tie2-CYP2C8, Tie2-sEH, and WT mice in the OIR model. They found that in OIR, there is increased NV in Tie2-CYP2C8 mice fed a diet with either ω-3 or ω-6 PUFAs, whereas there is reduced NV in Tie2-sEH mice. These results support the hypothesis that both of ω-3 and ω-6 PUFA lipid products of CYP2C promote NV in the retina and advanced DR. Recently, an exciting study by Fleming and colleagues [59] has demonstrated that sEH is involved in the metabolism of 19, 20-epoxydocosapentaenoic acid (19, 20-EDP; DHA-CYP-derived product) to 19,20-dihydroxydocosapentaenoic acid (19, 20-DHDP). They also showed that the accumulation of 19, 20-DHDP, and overexpression of sEH in the retinal Müller glial cells causes retinopathy [59]. Although these findings support the importance of sEH and 19, 20-DHDP in the Müller glial cells, the same research group (Fleming and colleagues) [127,128] has shown that increasing EETs levels in the retinal endothelial cells causes retinopathy, which supports our hypothesis indicating in Figure 5. These studies [59,105] suggest that regarding CYP-mediated ω-3 PUFA products in angiogenesis could be cell- or tissue-dependent.

## 5. Conclusions and Perspectives

It is well established that PGs play a vital role in the regulation of retinal blood flow and choroidal blood flow. Overproduction of TXA2 and PGE2 promotes ROS formation, which is associated with cardiovascular diseases. COX blockade decreases retinal VEGF and inhibits NF-κB activation, which regulates the production of inflammatory mediators and ROS production in the retina. Clinical studies suggest that high dose of aspirin might have beneficial effects in the development of DR in patients with early stages of DR. Since PGE2 promotes the production of retinal VEGF via EP4 receptor, the development of novel pharmacological agents targeting EP4 receptors could be an important area for clinical studies in patients with DR.

12-HETE, one of the LOX-derived eicosanoids, is a potent vasodilator, whereas 15-HETE causes vasoconstriction. Deletion of the 5-LOX gene decreases capillary degeneration, leukostasis, and activation of NF-κB in diabetic animals, suggesting that 5-LOX-derived proinflammatory metabolites play a role in the pathogenesis of DR. There is a significant elevation of 12-HETE and 15-HETE levels in the vitreous of diabetic patients with PDR. There is some evidence that 12-HETE or 15-HETE generates ROS through NOX, creating a status of oxidative stress, which is an important factor in the development of DR. More research is needed to determine whether 12/15-LOX blockade can prevent the development of DR in clinical studies.

EETs and 20-HETE, the CYP-derived eicosanoids, are produced in the retinal circulation. There is some evidence that glial stimulation elicited vasodilation via EETs, whereas glial stimulation results in vasoconstriction mediated by 20-HETE in the retina. Less research is known about the role of EETs in the development of DR. Several studies indicate that EETs are implicated in retinal angiogenesis, especially under hypoxia conditions. RAS activation contributes to retinal hyperpermeability and retinal neovascularization in DR. Although the blockade of RAS with an AT1 receptor blocker has reduced the incidence of DR in clinical studies, the AT1 blockade did not reduce the progression of DR in diabetic patients. Our recent publication [64] shows that in DR, RAS activation augments retinal levels of sEH, degrading EETs to compensate RAS-induced retinal microvascular damage. In the future study, it is needed to test whether EETs or EET receptor (Figure 2) blockade can optimize RAS blockade to prevent or reduce DR-induced microvascular damage.

Growing evidence support that ω-3 PUFA metabolites via COX- and LOX-pathway inhibit inflammation and retinal angiogenesis. On the other hand, the ω-3 and ω-6 PUFA metabolites via CYP2C promote neovascularization in the retina. Thus, more research is needed to determine whether ω-3 PUFA supplementation along with the CYP2C blockade can prevent the development of advanced DR.

In conclusion, although anti-VEGF and photocoagulation therapy have improved care in DR patients, these therapies still cause significant side effects. Thus, there is a critical need to identify new therapeutic targets to treat or prevent DR progression. COX-derived, LOX-derived, and CYP-derived eicosanoids regulate oxidative stress, retinal VEGF levels, and inflammatory cytokines, as well as being involved in the pathophysiology of DR. There are many available approaches, including selective blockers, transgenic mice, and knockout mice, to permit researchers to study the functions of these lipid mediators in the retina as well as their role in the pathophysiology of DR. These lipid mediators are essential therapeutic targets for DR because many receptors and these three eicosanoid pathways are well defined. Thus, we propose that COX blockers (aspirin and celecoxib), EET blockers (montelukast (a selective CYP2C inhibitor [129,130]) and DC260126 (an EET receptor antagonist [131,132])), and 12/15-LOX blockers (baicalein, NDGA, and CDC) could be the potential therapeutic methods to prevent the development of early-stage DR (Figure 6). We also propose that EP4 receptor blocker (L-161982), ω-3 PUFA diet + EET blockers (montelukast and DC260126), AT1 blockers + EET blockers (montelukast and DC260126), and 12/15-LOX blockers (baicalein, NDGA, and CDC) could be the potential therapeutic methods to prevent the development of proliferative DR (Figure 6). Finally, since oxidative stress represents a vital regulator of eicosanoids-mediated microvascular dysfunction in DR and other diseases, antioxidants could be a therapeutic approach to interrupt eicosanoid signaling and in turn improving visual outcome in DR.

## Figures and Tables

**Figure 1 antioxidants-09-00520-f001:**
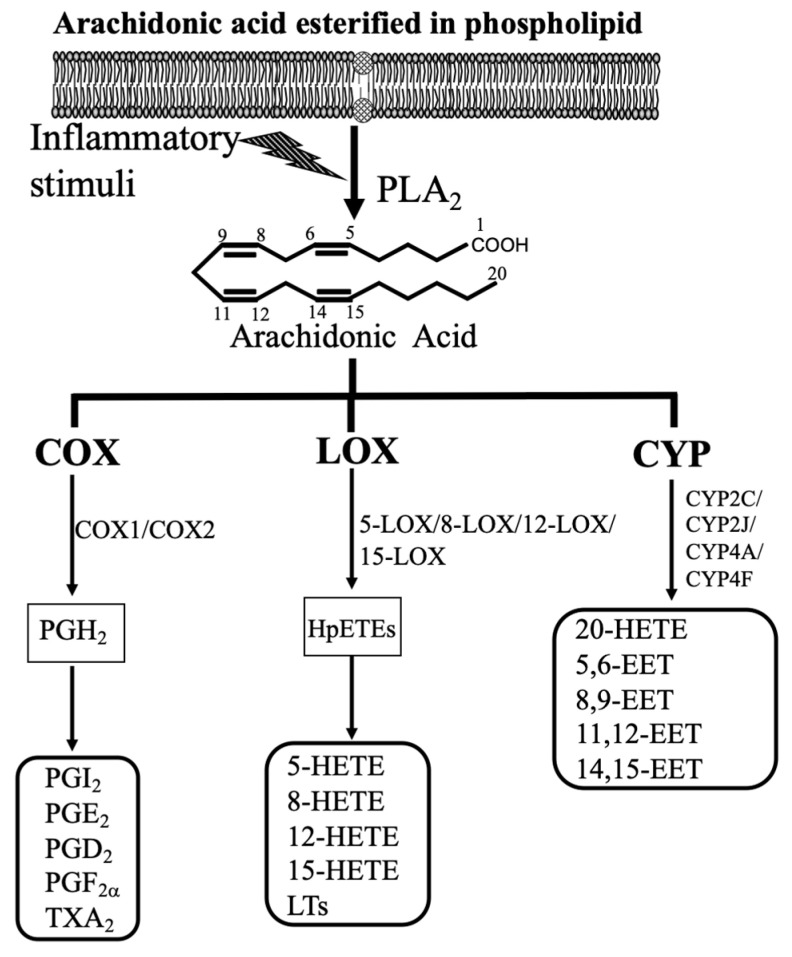
Bioactive eicosanoids derived from arachidonic acid (AA) cascade. After trigging by inflammatory conditions such as the presence of cytokines and growth factors, AA-containing phospholipids are hydrolyzed by phospholipase A2 (PLA2), resulting in the release of free AA. AA can be further metabolized by three pathways, i.e., the cyclooxygenase (COX), lipoxygenase (LOX), and cytochrome P450 (CYP) pathways. AA cascade generates prostaglandins (PGs), thromboxane A2 (TXA2), and a series of hydroxyeicosatetraenoic acids (HETEs), leukotrienes (LTs), and epoxyeicosatrienoic acids (EETs).

**Figure 2 antioxidants-09-00520-f002:**
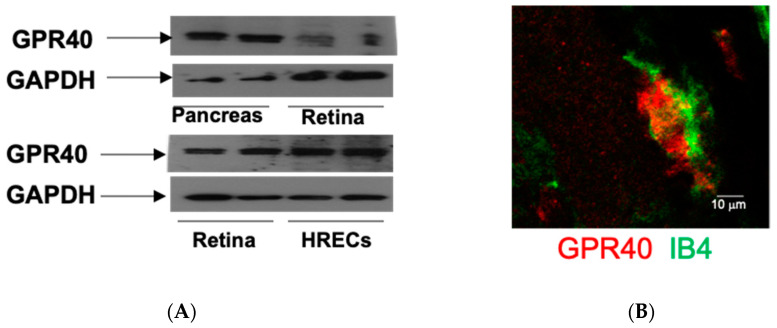
(**A**) EET receptor (GPR40) protein is expressed in the retina and human retinal endothelial cells (HRECs). The samples from the pancreas were used as a positive control. (**B**) GPR40 (red) and IB4 (green; a marker of the retinal blood vessel) in the retina. The retinal and pancreatic samples were isolated from male mice.

**Figure 3 antioxidants-09-00520-f003:**
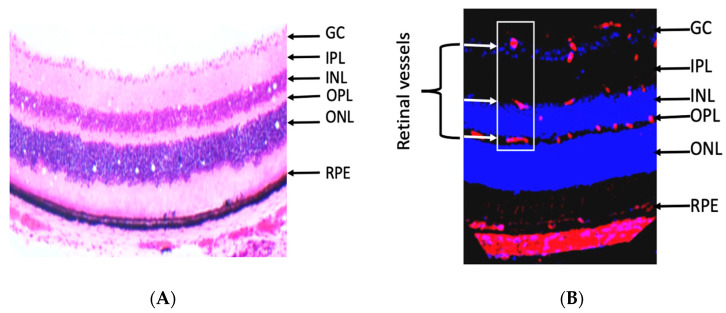
(**A**) Retinal section, which was isolated from mice, stained with hematoxylin and eosin showing different retinal layers as seen under the microscope from inside to outside are the ganglion cell (GC), inner plexiform (IPL), inner nuclear (INL), outer plexiform (OPL), outer nuclear (ONL), and retinal pigment epithelium (RPE). (**B**) Immunostaining of retinal section (mouse) with the vascular marker (isolectin-B4, red) and nuclear marker (DAPI, blue), showing that retinal blood vessels (inside the white box) are primarily localized in the inner retinal layers (the nerve fiber and ganglion cell layer and plexiform layer). The arrow in B is pointing to inner retinal layers that contain blood vessels (red).

**Figure 4 antioxidants-09-00520-f004:**
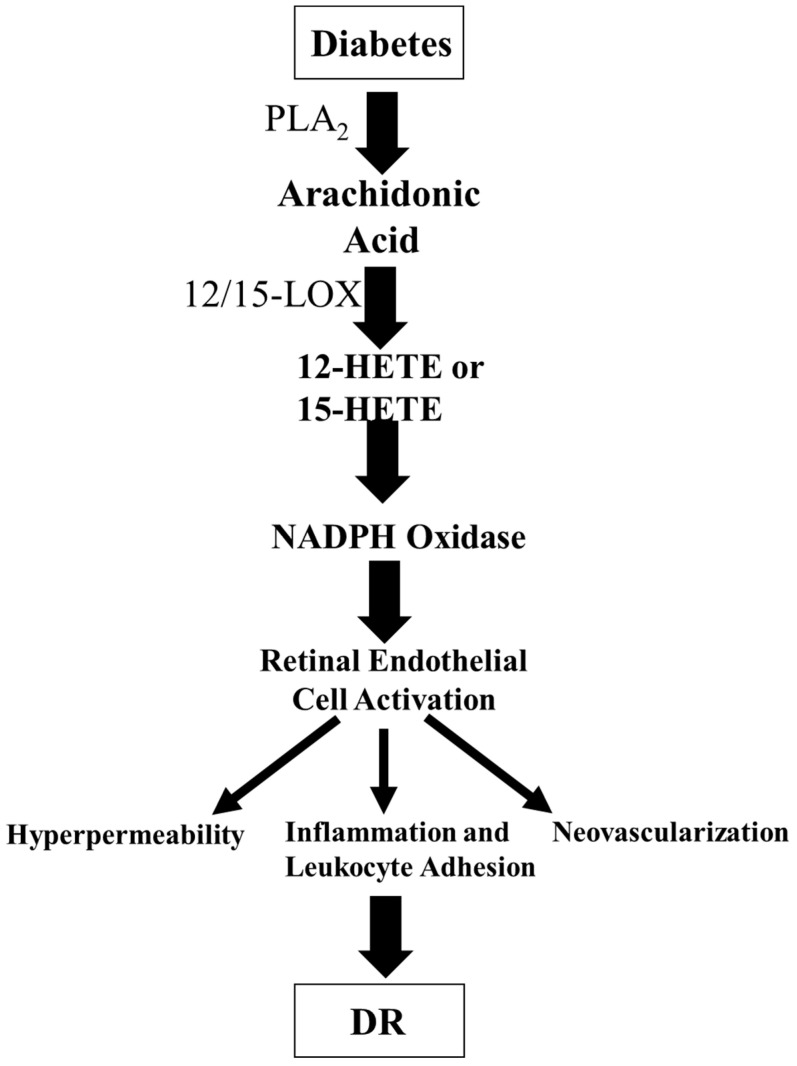
Cascade events involved in the pathogenesis of DR: Hyperglycemia activates the PLA2 to release AA from the retinal cell membrane. AA is converted to 12- or 15-HETE that generates ROS through NOX, creating a status of oxidative stress. This oxidative stress leads to the activation of retinal endothelial cells through various inflammatory signaling pathways, leading to leukocyte adhesion, hyperpermeability, and ultimately NV (the cardinal signs of DR).

**Figure 5 antioxidants-09-00520-f005:**
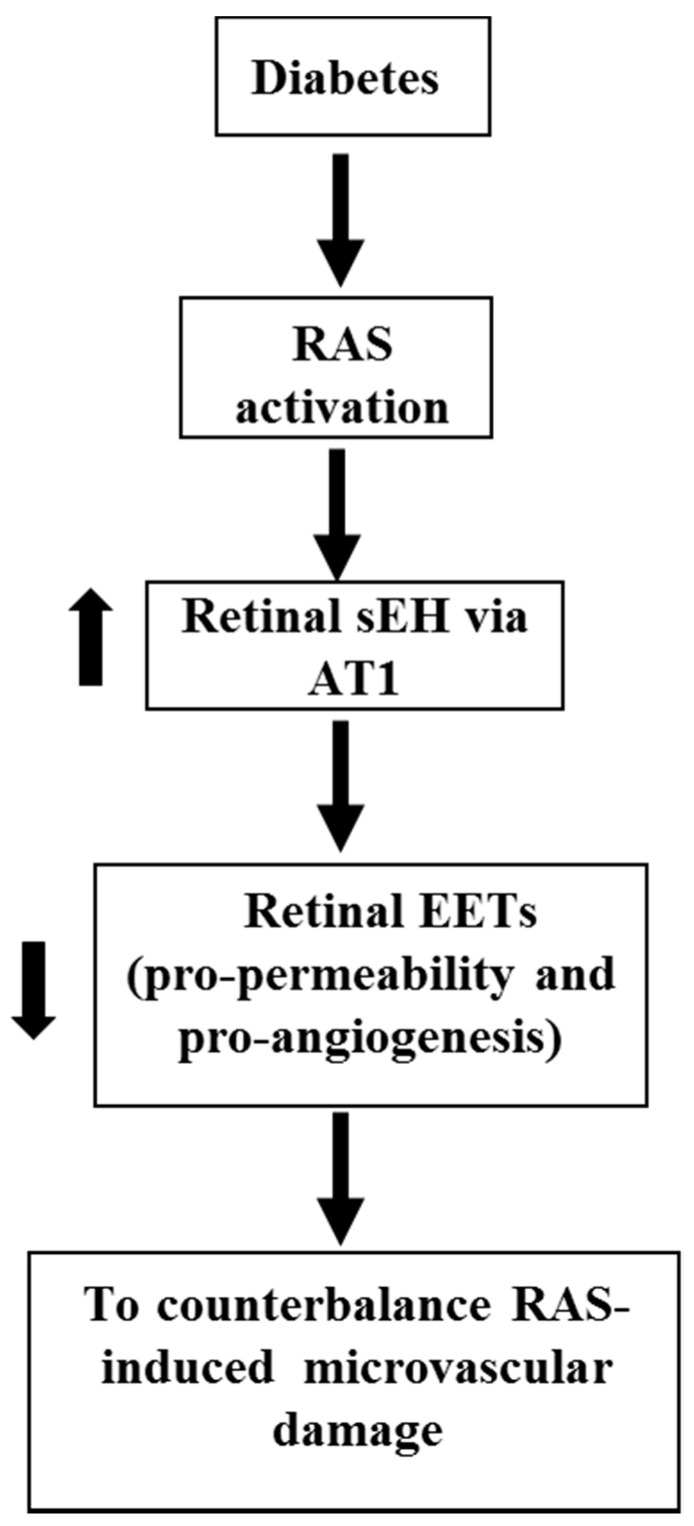
The hypothesis about the interaction of sEH/EETs and RAS in DR. During diabetes, RAS activation augments retinal sEH, via AT1, which decreases EETs (pro-permeability and pro-angiogenesis factors) to counterbalance the effects of RAS on retinal microvascular damage.

**Figure 6 antioxidants-09-00520-f006:**
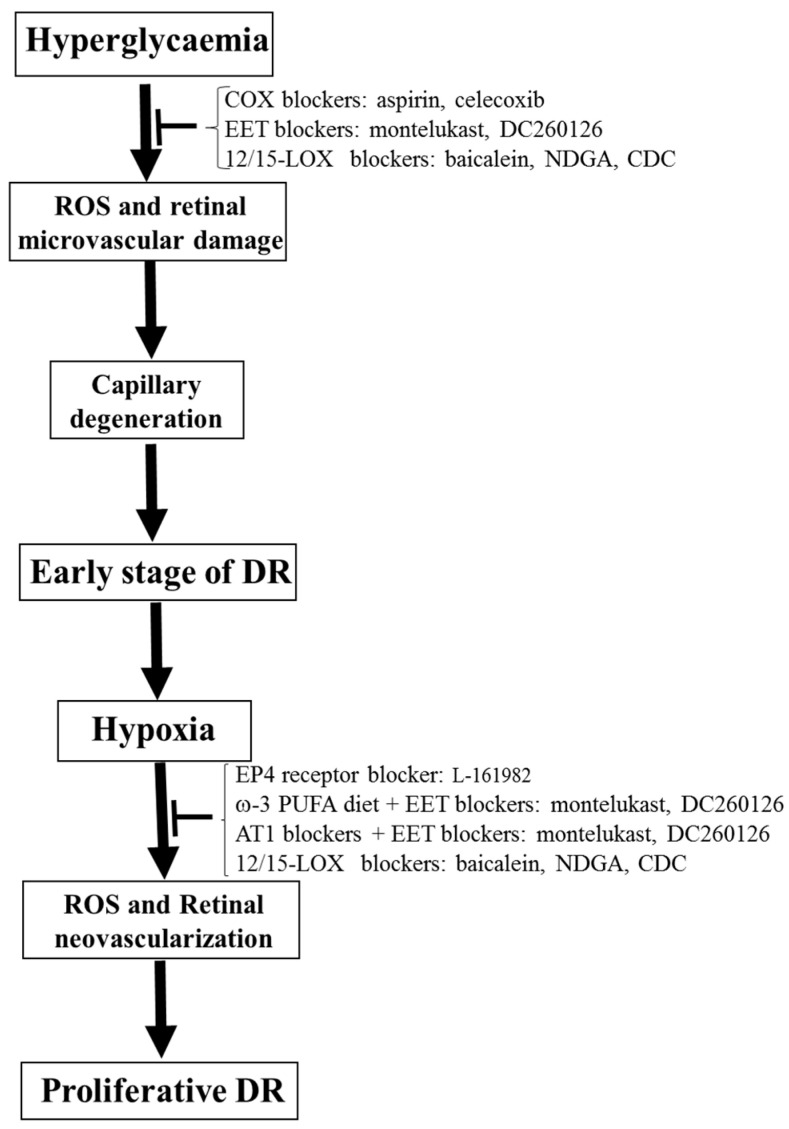
General mechanisms for the development of the early stage of DR and proliferative DR. We also indicate the potential therapeutic methods of the eicosanoid pathways to prevent the development of the early stage of DR and proliferative DR.

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
