# Peer review of "Eicosanoids and Oxidative Stress in Diabetic Retinopathy"

_antioxidants, 2020, doi:10.3390/antiox9060520_

Round 1

Reviewer 1 Report

the paper is speculative, but it is difficult to recognize a clinical pattern of experiments, it is interesting, bun needs a better organization in explaining concepts in a readible way

Author Response

We appreciate the reviewers’ careful reading and thoughtful comments, and the changes are shown in “underline” in the revised manuscript.  Our responses follow.

Reviewer 1:

The paper is speculative, but it is difficult to recognize a clinical pattern of experiments, it is interesting, but needs a better organization in explaining concepts in a readible way.

            According to the reviewer’s comment, we have made significant modification to obtain better organization in the revised manuscript.

Reviewer 2 Report

Summary:

The present study reflects the authors' efforts to analyse and summarise the data of a remarkable number of publications regarding on the pathophysiology of diabetic retinopathy, and the role of eicosanoids and oxidative stress in the development and progression of the disease. The work could serve as a helpful material for understanding the function eicosanoids and the pathomechanism of diabetic retinopathy.

The authors present their results in a logical and meaningful manner, nevertheless some revision are required before publication.

  1. In the “Introduction” section, line 91: I think the authors probably have wanted to write PGs instead of PG2.
  2. Also in this section, the authors describe the synthesis of LOX-derived eicosanoids, however about 5-HETE, 8-HETE synthesis pathways and their functions the authors have not discuss.
  3. Lines 339 and 428: it is recommended to use post-STZ injection or post-STZ treatment instead of the current phrasing.
  4. Lines 116, 148 and 437: the meaning of the abbreviations (VSMCs, NOX and Ang II) is not provided.
  5. In chapter 4.1 the authors discuss about the intervention and effect of aspirin in some signal pathways of eicosanoids in diabetic retinopathy. In the mentioned clinical studies, 650 and 990 mg daily dose of aspirin were used, which did not bring the expected results, while Zheng et al. found aspirin to be effective in animal experiments. Please, provide the dose of aspirin used by Zheng et al. [6], and explain the discrepancies between these results. And finally, regarding this topic, what think the authors about the use of a lower aspirin dose in the prevention and/or treatment of DR?
  6. The “Conclusion” section is too long, resembling a “Discussion” rather than a “Conclusion” section. Please, revise and rephrase this section.

Author Response

We appreciate the reviewers’ careful reading and thoughtful comments, and the changes are shown in “underline” in the revised manuscript.  Our responses follow.

Reviewer 2:

  1. In the “Introduction” section, line 91: I think the authors probably have wanted to write PGs instead of PG2.

        We have made the correction.

  1. Also in this section, the authors describe the synthesis of LOX-derived eicosanoids, however about 5-HETE, 8-HETE synthesis pathways and their functions the authors have not discuss.

       We have discussed the synthesis and function of 5-HETE and 8-HETE        (between Line 127 to Line 129 of the revised manuscript).

  1. Lines 339 and 428: it is recommended to use post-STZ injection or post-STZ treatment instead of the current phrasing.

        Following the reviewer’s comment, we have made the correction.

  1. Lines 116, 148 and 437: the meaning of the abbreviations (VSMCs, NOX and Ang II) is not provided.

        We have made the correction.

  1. In chapter 4.1 the authors discuss about the intervention and effect of aspirin in some signal pathways of eicosanoids in diabetic retinopathy. In the mentioned clinical studies, 650 and 990 mg daily dose of aspirin were used, which did not bring the expected results, while Zheng et al. found aspirin to be effective in animal experiments. Please, provide the dose of aspirin used by Zheng et al. [6], and explain the discrepancies between these results. And finally, regarding this topic, what think the authors about the use of a lower aspirin dose in the prevention and/or treatment of DR?

        We agree with the reviewer’s comment.  We have discussed the dose of aspirin used in the previous paper by Zheng et al., and explain the results between this animal study and clinical studies in DR treatment (Line 316-Line 320 of the revised manuscript).

  1. The “Conclusion” section is too long, resembling a “Discussion” rather than a “Conclusion” section. Please, revise and rephrase this section.

        Following the reviewer’s comment, we have shortened and rephrased the “Conclusion” section.

Reviewer 3 Report

The review titled Eicosanoids and Oxidative Stress in Diabetic Retinopathy submitted by Wang, et al., is well written. Eicosanoid biological functions in the retinal system and diabetic retinopathy are extensively covered. The topics are organized conveniently to navigate readers along eicosanoid pathways, including alternations that occur in eicosanoid pathways  in diabetic retinopathy.

The review should provide more information on data in Figure 2 and Figure 3. It is not clear whether the data and images are original or borrowed from a previous study. If the latter, then permissions should be indicated in figure legends. What is the significance of the arrows in Fig. 2 B? Arrows are not indicated in discussions or the legends.

The introduction section should be effectually edited. The introduction in its present state (particularly the first paragraph) is a bit frazzled and the flow of information is rough. Some sentences are choppy and could be improved considerably with a strong edit.

The main body of the review is stimulating, providing a good synopsis of the roles of COX, LOX, and CYP in the retinal system and diabetic retinopathy development.

The conclusion section effectively tethers findings from eicosanoid studies in the scientific literature to potential development and use of various different eicosanoid blockers to prevent diabetic retinopathy development and progression.

Author Response

Reviewer 3:

  1. The review should provide more information on data in Figure 2 and Figure 3. It is not clear whether the data and images are original or borrowed from a previous study. If the latter, then permissions should be indicated in figure legends. What is the significance of the arrows in Fig. 2 B? Arrows are not indicated in discussions or the legends.

        The data and images of Figure 2 and Figure 3 are original.  Following the reviewer’s comment, we have deleted Arrows in Figure 2B.   We have modified the figure legends of Figure 2 and Figure 3.

  1. The introduction section should be effectually edited. The introduction in its present state (particularly the first paragraph) is a bit frazzled and the flow of information is rough. Some sentences are choppy and could be improved considerably with a strong edit.

        Following the reviewer’s comment, we have revised the paragraph of the “Introduction” section of the revised manuscript.

Round 2

Reviewer 1 Report

the paaper is correct and the AA have addressed some major concerns it would be published

Author Response

According to the reviewer’s comment, we have modified the discussion of clinical experiments in the COX-eicosanoids.  Also, we have done the spell check in the revised manuscript.

Reviewer 2 Report

Summary:

The manuscript was revised by authors and was improved considerably, according to the recommendations. However, there is one suggestion to the authors.

I reserve my opinion that the “Conclusion” section is too long, resembling a “Discussion” rather than a “Conclusion” section. My suggestion is that this section need to be rename in the case that the authors want to maintain this format, as well as to insert a short, concise Conclusion as a final part of the manuscript.

Author Response

    According to the reviewer’s comment, we have changed the final section from “Conclusion” to “Conclusions and Perspectives.”  Also, we have added a short and concise conclusion in the final part of the revised manuscript.